# MOTIF-AWARE ATTRIBUTE MASKING FOR MOLECULAR GRAPH PRE-TRAINING

## ABSTRACT

Attribute reconstruction is used to predict node or edge features in the pre-training of graph neural networks. Given a large number of molecules, they learn to capture structural knowledge, which is transferable for various downstream property prediction tasks and vital in chemistry, biomedicine, and material science. Previous strategies that randomly select nodes to do attribute masking leverage the information of local neighbors. However, the over-reliance of these neighbors inhibits the model's ability to learn long-range dependencies from higher-level substructures. For example, the model would learn little from predicting three carbon atoms in a benzene ring based on the other three but could learn more from the inter-connections between the functional groups, or called chemical motifs. To explicitly determine inter-motif knowledge transfer of pre-trained model, we define inter-motif node influence measures. Then, we propose and investigate motif-aware attribute masking strategies to capture long-range inter-motif structures by leveraging the information of atoms in neighboring motifs. Once each graph is decomposed into disjoint motifs, the features for every node within a sample motif are masked. The graph decoder then predicts the masked features of each node within the motif for reconstruction. We evaluate our approach on eight molecular property prediction datasets and demonstrate its advantages.

## 1 INTRODUCTION

Molecular property prediction has been an important topic of study in fields such as physical chemistry, physiology, and biophysics (Wu et al., 2017). It can be defined as a graph label prediction problem and addressed by machine learning. However, graph learning models such as graph neural networks (GNNs) must overcome issues in data scarcity, as the creation and testing of real-world molecules is an expensive endeavor (Chang et al., 2022). To address labeled data scarcity, model pre-training has been utilized as a fruitful strategy for improving a model's predictive performance on downstream tasks, as pre-training allows for the transfer of knowledge from large amounts of unlabeled data. The selection of pre-training strategy is still an open question, with contrastive tasks (Zhu et al., 2021) and predictive/generative tasks (Hu et al., 2020a) being the most popular methods.

Attribute reconstruction is one predictive method for graphs that utilizes masked autoencoders to predict node or edge features (Hu et al., 2020a; Kipf & Welling, 2016; Xia et al., 2022). Masked autoencoders have found success in vision and language domains (He et al., 2022; Devlin et al., 2018) and have been adopted as a pre-training objective for graphs as the reconstruction task is able to transfer structural pattern knowledge (Hu et al., 2020a), which is vital for learning specific domain knowledge such as valency in material science. Additional domain knowledge which is important for molecular property prediction is that of functional groups, also called chemical motifs (Pope et al., 2019). *The presence and interactions between chemical motifs directly influence molecular properties, such as reactivity and solubility* (Frechet, 1994; Plaza et al., 2014). Prior work in message passing for quantum chemistry has shown that long-range dependencies are important for downstream prediction in chemical domains (Gilmer et al., 2017). Therefore, to capture the interaction information between motifs, it is important to transfer inter-motif structural knowledge and other long-range dependencies during the pre-training of graph neural networks.

Unfortunately, the random attribute masking strategies used in previous work for graph pre-training were not able to capture the long-range dependencies inherent in inter-motif knowledge (Kipf &

**Random Attribute Masking**
e.g., AttrPred (Hu et al. 2020), GraphMAE (Hou et al. 2022)

**Motif-aware Attribute Masking**
ours, named MoAMa

- 🔵 attribute-observed nodes
- ⚫ attribute-masked nodes
- ↻ intra-motif structural information passing
- ⟲ inter-motif feature information passing

Figure 1: Our MoAMa masks every node in sampled motifs to pre-train GNNs. The full masking of a motif forces the GNNs to learn to (1) pass feature information across motifs and (2) pass local structural information within the motif. Compared to the traditional random attribute masking strategies, the motif-aware masking captures the most essential information to learn graph embeddings. Random masking would put most of the pre-training effort on passing the feature information within a motif, e.g., predicting two carbon nodes in a benzene ring based on the other four.

Welling, 2016; Hu et al., 2020b; Pan et al., 2019). That is because they rely on neighboring node feature information for reconstruction (Hu et al., 2020a; Hou et al., 2022). Notably, leveraging the features of local neighbors can contribute to learning important local information, including valency and atomic bonding. However, GNNs heavily rely on the neighboring node's features rather than graph structure (Yun et al., 2021), and this over-reliance inhibits the model's ability to learn from motif structures as message aggregation will prioritize local node feature information due to the propagation bottleneck (Alon & Yahav, 2021). For example, as shown on the left-hand side of Figure 1, if only a (small) partial set of nodes were masked in several motifs, the pre-trained GNNs would learn to predict the node types (i.e., carbon) of two atoms in the benzene ring based on the features and structure of the other four carbon atoms in the ring, limiting the knowledge transfer of long-range dependencies. To measure the inter-motif knowledge transfer of graph pre-training strategies, we define five inter-motif influence measurements and report our findings in Sec. 6.

Recent successes in vision and language domains have shown the utility of masking semantically related regions, such as pixel batches (Li et al., 2022; Xie et al., 2022; He et al., 2021) and multi-token spans (Levine et al., 2020; Sun et al., 2019; Joshi et al., 2020), and have demonstrated that a random masking strategy is not guaranteed to transfer necessary inter-part relations and intra-part patterns (Li et al., 2022). To better enable the transfer of long-range inter-part relations downstream, we propose a novel semantically-guided masking strategy based on chemical motifs. In Figure 1, we visually demonstrate our method for motif-aware attribute masking, where each molecular graph is decomposed into disjoint motifs. Then the node features for each node within the motif will be masked by a mask token. A graph decoder will predict the masked features of each node within the motif as the reconstruction task. The benefits of this strategy are twofold. First, because all features of the nodes within the motif are masked, our strategy reduces the amount of feature information being passed within the motif and relieves the propagation bottleneck, allowing for the greater transfer of inter-motif feature and structural information. Second, the masking of all intra-motif node features explicitly forces the decoder to transfer intra-motif structural information. A novel graph pre-training solution based on the **Mo**tif-aware **A**ttribute **Ma**sking strategy, called **MoAMa**, is able to learn long-range inter-motif dependencies with knowledge of intra-motif structure. We evaluate our strategy on eight molecular property prediction datasets and demonstrate its improvement to inter-motif knowledge transfer as compared to previous strategies.

## 2 RELATED WORK

**Molecular graph pre-training** The prediction of molecular properties based on graphs is important (Wu et al., 2017). Molecules are scientific data that are time- and computation-intensive to collect and annotate for different property prediction tasks (Liu et al., 2023). Many self-supervised learning methods (Hu et al., 2020a; Hou et al., 2022; Zhang et al., 2021; Kim et al., 2022; Xia et al., 2023) were proposed to capture the transferable knowledge from another large scale of molecules without annotations. For example, AttrMask (Hu et al., 2020a) randomly masked atom attributes for

prediction. GraphMAE (Hou et al., 2022) pre-trained the prediction model with generative tasks to reconstruct node and edge attributes. D-SLA (Kim et al., 2022) used contrastive learning based on graph edit distance. These pre-training tasks could not well capture useful knowledge for various domain-specific tasks since they fail to incorporate important domain knowledge in pre-training. A great line of prior work (Zhang et al., 2021; Rong et al., 2020; Sun et al., 2021) used graph motifs which are the recurrent and statistically significant subgraphs to characterize the domain knowledge contained in molecular graph structures, e.g., functional groups. However, their solutions were tailored to specific frameworks for either generation-based or contrast-based molecular pre-training. Additionally, explicit motif type generation/prediction inherently does not transfer intra-motif structural information and is computationally expensive due to the large number of prediction classes. In this work, we study on the strategies of attribute masking with the awareness of domain knowledge (i.e., motifs), which plays an essential role in self-supervised learning frameworks (Xia et al., 2023).

**Masking strategies on molecules**   Attribute masking of atom nodes is a popular method in graph pre-training given its broad usage in predictive, generative, and contrastive self-supervised tasks (Hu et al., 2020a;b; Hou et al., 2022; You et al., 2020; 2021). For example, predictive and generative pre-training tasks (Hu et al., 2020a; Hou et al., 2022; Xia et al., 2023) mask atom attributes for prediction and reconstruction. Contrastive pre-training tasks (You et al., 2020; 2021) mask nodes to create another data view for alignment. Despite the widespread use of attribute masking in molecular pre-training, there is a notable absence of comprehensive research on its strategy and effectiveness. Previous studies have largely adopted strategies from the vision and language domains (He et al., 2022; Devlin et al., 2018), where atom attributes are randomly masked with a predetermined ratio. Since molecules are atoms held together by strict chemical rules, the data modality of molecular graphs is essentially different from natural images and languages. For molecular graphs, random attribute masking results in either over-reliance on intra-motif neighbors (Dwivedi et al., 2023) or breaking the inter-motif connections via random edge masking. In this work, we introduce a novel strategy of attribute masking, which turns out to capture and transfer useful knowledge from intra-motif structures and long-range inter-motif node features.

## 3 PRELIMINARIES

**Graph property prediction**   Given a graph $G = (\mathcal{V}, \mathcal{E}) \in \mathcal{G}$ with the node set $\mathcal{V}$ for atoms and the edge set $\mathcal{E} \subset \mathcal{V} \times \mathcal{V}$ for bonds, we have a $d$-dimensional node attribute matrix $\mathbf{X} \in \mathbb{R}^{|\mathcal{V}| \times d}$ that represents atom features such as atom type and chirality. We use $y \in \mathcal{Y}$ as the graph-level property label for $G$, where $\mathcal{Y}$ represents the label space. For graph property prediction, a predictor with the encoder-decoder architecture is trained to encode $G$ into a representation vector in the latent space and decode the representation to predict $\hat{y}$. The training process optimizes the parameters to make $\hat{y}$ to be the same as the true label value $y$. A GNN is a commonly used encoder that generates $k$-dimensional node representation vectors, denoted as $\mathbf{h}_v \in \mathbb{R}^k$, for any node $v \in \mathcal{V}$:

$$\mathbf{H} = \{\mathbf{h}_v : v \in \mathcal{V}\} = \text{GNN}(G) \in \mathbb{R}^{|\mathcal{V}| \times k}. \tag{1}$$

Here $\mathbf{H}$ is the node representation matrix for the graph $G$. Without loss of generality, we implement Graph Isomorphism Networks (GIN) (Xu et al., 2019) as the choice of GNN in accordance with previous work (Hu et al., 2020a). Once the set of node representations are created, a $\text{READOUT}(\cdot)$ function (such as max, mean, or sum) is used to summarize the node-level representation into graph-level representation $\mathbf{h}_G$ for any $G$:

$$\mathbf{h}_G = \text{READOUT}(\mathbf{H}) \in \mathbb{R}^k. \tag{2}$$

The graph-level representation vector $\mathbf{h}_G$ is subsequently passed through a multi-layer perceptron (MLP) to generate the label prediction $\hat{y}$, which exists in the label space $\mathcal{Y}$:

$$\hat{y} = \text{MLP}(\mathbf{h}_G) \in \mathcal{Y}. \tag{3}$$

**GNN pre-training**   Random initialization of the predictor's parameters would easily result in suboptimal solutions for graph property prediction. This is because the number of labeled graphs is usually small. It prevents a proper coverage of task-specific graph and label spaces (Hu et al., 2020a; Liu et al., 2023). To improve generalization, GNN pre-training is often used to warm-up the model parameters based on a much larger set of molecules without labels. In this work, we focus on the attribute masking strategy for GNN pre-training that aims to predict the masked values of node attributes given the unlabeled graphs.

## 4 INTER-MOTIF INFLUENCE

To measure the influence generally from (either intra-motif or inter-motif) source nodes on a target node $v$, we design a measure that quantifies the influence from any source node $u$ in the same graph $G$, denoted by $s(u, v)$. $\mathbf{h}_v$ was learned by Eq. (1) and was influenced by node $u$. When the embedding of $u$ is eliminated from GNN initialization, i.e., set $\mathbf{h}_u^{(0)} = \vec{0}$, Eq. (1) would produce a new representation vector of node $v$, denoted by $\mathbf{h}_{v,\text{w/o } u}$. We use the $L^2$-norm to define the influence:

$$s(u, v) = \|\mathbf{h}_v - \mathbf{h}_{v,\text{w/o } u}\|_2. \tag{4}$$

The collective influence from a group of nodes in a motif $M = (\mathcal{V}_M, \mathcal{E}_M)$ is measured as follows:

$$s_{\text{motif}}(v, M) = \frac{1}{|\mathcal{V}_M \setminus \{v\}|} \sum_{u \in \mathcal{V}_M \setminus \{v\}} s(u, v). \tag{5}$$

Suppose the target node $v$ is in the motif $M_v = (\mathcal{V}_{M_v}, \mathcal{E}_{M_v})$. Using $M_v$ as the target motif, the influence from intra-motif and inter-motif nodes can be calculated as:

$$s_{\text{intra}}(v) = s_{\text{motif}}(v, M_v); \; s_{\text{inter}}(v) = \frac{\sum_{M \in \mathcal{M} \setminus \{M_v\}} |\mathcal{V}_M| \times s_{\text{motif}}(v, M)}{|\mathcal{V} \setminus \mathcal{V}_{M_v}|}. \tag{6}$$

Usually the number of inter-motif nodes is significantly bigger than the number of intra-motif nodes, i.e., $|\mathcal{V}| \gg |\mathcal{V}_{M_v}|$, which reveals two issues in the influence measurements. First, when the target motif is too small (e.g., has only one or two nodes), the intra-motif influence cannot be defined or is defined on the interaction with only one neighbor node. Second, most inter-motif nodes are not expected to have any influence, so the average function in Eq. (5) would lead comparisons to be biased to intra-motif influence. To address the two issues, we constrain the influence summation to be on the *same number* of nodes (i.e., top-$k$) from the intra-motif and inter-motif node groups. Explicitly, this means $u \in \mathcal{V}_M / \{v\}$ in Eq. (5) is sampled from the top-$k$ most influencial nodes (top-3). The ratio of inter- to intra-motif influence over the graph dataset $\mathcal{G}$ is then defined as:

$$\text{InfRatio}_{\text{node}} = \frac{1}{\sum_{(\mathcal{V}, \mathcal{E}) \in \mathcal{G}} |\mathcal{V}|} \sum_{(\mathcal{V}, \mathcal{E}) \in \mathcal{G}} \sum_{v \in \mathcal{V}} \frac{s_{\text{inter}}(v)}{s_{\text{intra}}(v)}, \tag{7}$$

$$\text{InfRatio}_{\text{graph}} = \frac{1}{|\mathcal{G}|} \sum_{G=(\mathcal{V}, \mathcal{E}) \in \mathcal{G}} \frac{1}{|\mathcal{V}|} \sum_{v \in \mathcal{V}} \frac{s_{\text{inter}}(v)}{s_{\text{intra}}(v)}, \tag{8}$$

where the average function is performed at the node level and graph level, respectively. Eq. (7) directly measures the influence ratios of all nodes $v$ within the dataset $\mathcal{G}$. However, this measure may include bias due to the distribution of nodes within each graph. We alleviate this bias in Eq. (8) by averaging influence ratios across each graph first.

While the InfRatio measurements are able to compare general inter- and intra-motif influences, these measures combine all inter-motif nodes into one set and do not consider the number of motifs in each graph. We further define rank-based measures that consider the distribution of motif counts across $\mathcal{G}$.

Let $\{M_1, ..., M_i, ..., M_n\}$ be an ordered set, where $M_i \in \mathcal{M}$ and $s_{\text{motif}}(v, M_i) \geq s_{\text{motif}}(v, M_j)$ if $i < j$. If $M_i = M_v$, we define $\text{rank}_v = i$. Note that graphs with only one motif are excluded as the distinction between inter and intra-motif nodes loses meaning. From this ranking, we define our score for inter-motif node influence averaged at the node, motif, and graph levels, derived from a similar score measurement used in information retrieval, Mean Reciprocal Rank (MRR) (Craswell, 2009):

$$\text{MRR}_{\text{node}} = \frac{1}{\sum_{(\mathcal{V}, \mathcal{E}) \in \mathcal{G}} |\mathcal{V}|} \sum_{(\mathcal{V}, \mathcal{E}) \in \mathcal{G}} \sum_{v \in \mathcal{V}} \frac{1}{\text{rank}_v}, \tag{9}$$

$$\text{MRR}_{\text{graph}} = \frac{1}{|\mathcal{G}|} \sum_{(\mathcal{V}, \mathcal{E}) \in \mathcal{G}} \frac{1}{|\mathcal{V}|} \sum_{v \in \mathcal{V}} \frac{1}{\text{rank}_v} \tag{10}$$

$$\text{MRR}_{\text{motif}} = \sum_{n=2}^{N} \frac{|\mathcal{G}^{(n)}|}{|\mathcal{G}| \sum_{(\mathcal{V}, \mathcal{E}) \in \mathcal{G}^{(n)}} |\mathcal{V}|} \sum_{(\mathcal{V}, \mathcal{E}) \in \mathcal{G}^{(n)}} \sum_{v \in \mathcal{V}} \frac{1}{\text{rank}_v}, \tag{11}$$

where $\mathcal{G}^{(n)} \subset \mathcal{G}$ is the set of graphs that contain $n \in [2, ..., N]$ motifs.

Similar to the InfRatio measurements, $\text{MRR}_{\text{node}}$ directly captures the impact of the influence ranks for each node within the full graph set, whereas $\text{MRR}_{\text{graph}}$ alleviates bias on the number of nodes within a graph by averaging across individual graphs first. Because these rank-based measurements are intrinsically dependent on the number of motifs within each graph, we additionally define $\text{MRR}_{\text{motif}}$ which weights the measurement towards popular motif counts within the data distribution. In information retrieval, MRR scores are used to quantify how well a system can return the most relevant item for a given query. Higher MRR scores indicate that relevant items were returned at higher ranks for each query. However, as opposed to traditional MRR measurements, where a higher rank for the most relevant item indicates better performance, lower scores are preferred for our MRR measurements as lower intra-motif influence rank indicate greater inter-motif node influence.

In Sec 6, we show the inter-motif node influence measurements of previous pre-trained models.

## 5 PROPOSED SOLUTION

In this section, we present our novel solution named MoAMa for effectively pre-training graph neural networks on molecular data. We will give details about the strategy of motif-aware attribute masking and reconstruction. Each molecule $G$ will have some portion of their node masked according to domain knowledge based motifs. We replace the node attributes of all masked nodes with a special mask token. Then, the GNN in Eq. (1) encodes the masked graph to the node representation space, and an MLP reconstructs the atom types for the attribute masked molecule.

### 5.1 KNOWLEDGE-BASED MOTIF EXTRACTION

To leverage the expertise from the chemistry domain, we extract motifs for molecules using the BRICS (Breaking of Retrosynthetically Interesting Chemical Substructures) algorithm (Degen et al., 2008). This algorithm leverages chemical domain knowledge by creating 16 rules for decomposition, the rules of which define the bonds that should be cleaved from the molecule in order to create a multi-set of disjoint subgraphs. Two key strengths of the BRICS algorithm over a motif-mining strategy (Geng et al., 2023) is that no training is required and important structural features, such as rings, are inherently preserved.

For each graph $G$, the BRICS algorithm decomposes the full graph into separate motifs. We denote the decomposition result as $\mathcal{M}_G = \{M_1, M_2, ..., M_n\}$, which is a set of $n$ motifs. Each motif $M_i = (\mathcal{V}_i, \mathcal{E}_i)$, for $i \in \{1, 2, ..., n\}$, is a disjoint subgraph of $G$ such that $\mathcal{V}_i \subset \mathcal{V}$ and $\mathcal{E}_i \subset \mathcal{E}$. For each motif multi-set $\mathcal{M}_G$, the union of all motifs $M_i \in \mathcal{M}_G$ should equal $G$. Formally, this means $\mathcal{V} = \bigcup_i V_i$ and $\mathcal{E} = (\bigcup_i E_i) \bigcup E_x$, where $E_x$ represents all the edges removed between motifs during the BRICS decomposition. Within the ZINC15 dataset (Sterling & Irwin, 2015), used for pre-training, each molecule has an average of 9.8 motifs, each of which have an average of 2.4 atoms.

### 5.2 MOTIF-AWARE ATTRIBUTE MASKING AND RECONSTRUCTION

To perform motif-aware attribute masking, $m$ motifs are sampled to form the multi-set $\mathcal{M}'_G \subset \mathcal{M}_G$ such that $(\sum_{(\mathcal{V}_i, \mathcal{E}_i) \in \mathcal{M}'_G} |\mathcal{V}_i|)/|\mathcal{V}| = \alpha$, for $\alpha$ is a chosen ratio value. The motifs sampled for $\mathcal{M}'_G$ must adhere to two criteria: (1) each node within the motif must be within a $k$-hop neighborhood ($k$ equals number of GNN layers) of an inter-motif node, and (2) sampled motifs may not be adjacent. These two criteria guarantee inter-motif knowledge access for each masked node. To adhere to the above criteria and account for variable motif sizes, we allow for some flexibility in the value for $\alpha$. We choose the bounds $0.15 < \alpha < 0.25$ in accordance to those used in previous works ($\alpha = 0.15$ (Hu et al., 2020a) and $\alpha = 0.25$ (Hou et al., 2022)).

Given a selected motif $M \in \mathcal{M}'_G$, nodes within $M$ have their attributes masked by replacing them with a mask token [MASK], which is a vector $\mathbf{m} \in \mathbb{R}^d$. Each element in $\mathbf{m}$ is a special value that is not present within the attribute space for that particular dimension. For example, we may set the attribute for the atom type dimension in $\mathbf{m}$ to the value 119, as we totally have 118 atom types (Hu et al., 2020a). We use $\mathcal{V}_{[\text{MASK}]} = \{v \in \mathcal{V}_i : M_i = (\mathcal{V}_i, \mathcal{E}_i) \in \mathcal{M}'_G\}$ to denote the set of all the masked nodes. We then define the input node features in the masked attribute matrix $\mathbf{X}_{[\text{MASK}]} \in \mathbb{R}^{|\mathcal{V}| \times d}$ for any $v \in \mathcal{V}$ using the following equation:

$$(\mathbf{X}_{[\text{MASK}]})_v = \begin{cases} \mathbf{X}_v, & v \notin \mathcal{V}_{[\text{MASK}]}, \\ \mathbf{m}, & v \in \mathcal{V}_{[\text{MASK}]}, \end{cases} \tag{12}$$

where $(\mathbf{X}_{\text{[MASK]}})_v$ and $\mathbf{X}_v$ denote the row of the node $v$ in $\mathbf{X}_{\text{[MASK]}}$ and $\mathbf{X}$, respectively. With a GNN encoder, all nodes with attributes $\mathbf{X}_{\text{[MASK]}}$ for the masked graph $G_{\text{[MASK]}}$ are encoded to the latent representation space according to Eq. (1): $\mathbf{H} = \text{GNN}(G_{\text{[MASK]}})$. $\mathbf{H}$ is then used to define the reconstruction loss of the node attributes:

$$\mathcal{L}_{\text{rec}} = \mathbb{E}_{v \in \mathcal{V}_{\text{[MASK]}}}[\log p(\mathbf{X}|\mathbf{H})], \tag{13}$$

where $p(\mathbf{X}|\mathbf{H})$ for the reconstruction attribute value is inferred by a decoder. In practice, reconstruction loss is measured using the scaled cosine error (SCE) (Hou et al., 2022), which calculates the difference between the probability distribution for the reconstruction attributes and the one-hot encoded target label vector. This choice of reconstruction loss is further discussed in later sections.

### 5.3 Design Space of the Attribute Masking Strategy

The design space of the motif-aware node attribute masking includes the following four parts:

**Masking distribution** We investigate the influence of masking distribution to the masking strategy using two factors to control the distribution of masked attributes:

- Percentage of nodes within a motif selected for masking: we propose to mask nodes from the selected motifs at different percentages. The percentage indicates the strength of the masked domain knowledge, which affects the hardness of the pre-training task of the attribute reconstruction.
- Dimension of the attributes: We propose to conduct either node-wise or element-wise (dimension-wise) masking. Element-wise masking selects different nodes for masking in different dimensions according to the percentage, while node-wise masking selects different nodes for all-dimensional attribute masking in different motifs.

**Reconstruction target** Existing molecular graph pre-training methods heavily rely on two atom attributes: atom type and chirality. Therefore, the reconstructive task could include one or both attributes using one or two different decoders. Experiments will find the most effective task definition.

**Reconstruction loss** We study different implementations of reconstruction loss functions for $\mathcal{L}_{\text{rec}}$. They include cross entropy (CE), scaled cosine error (SCE) (Hou et al., 2022), and mean square error (MSE). GraphMAE (Hou et al., 2022) suggested that SCE was the best loss function, however, it is worth investigating the effect of the loss function choices in the motif-based study.

Additionally, attribute masking focuses on local graph structures and suffers from representation collapse (Hu et al., 2020a; Hou et al., 2022). To address this issue, we use a knowledge-enhanced auxiliary loss $\mathcal{L}_{\text{aux}}$ to complement $\mathcal{L}_{\text{rec}}$. Given any two graphs $G_i$ and $G_j$ from the graph-based chemical space $\mathcal{G}$, $\mathcal{L}_{\text{aux}}$ first calculates the Tanimoto similarity (Bajusz et al., 2015) between $G_i$ and $G_j$ as $\text{Tanimoto}(G_i, G_j)$ based on the bit-wise fingerprints, which characterizes frequent fragments in the molecular graphs. Then $\mathcal{L}_{\text{aux}}$ aligns the latent representations with the Tanimoto similarity using the cosine similarity, inspired by previous work (Atsango et al., 2022). Formally, we define:

$$\mathcal{L}_{\text{aux}} = \sum_{i,j} \left( \text{Tanimoto}(G_i, G_j) - \text{cosine}(\mathbf{h}_{G_i}, \mathbf{h}_{G_j}) \right), 1 \leq i, j \leq |\mathcal{G}|, i \neq j, \tag{14}$$

where $\mathbf{h}_{G_i}$ and $\mathbf{h}_{G_j}$ are the graph representation of $G_i$ and $G_j$, respectively. The full pre-training loss is $\mathcal{L} = \beta \mathcal{L}_{\text{rec}} + (1 - \beta) \mathcal{L}_{\text{aux}}$, where $\beta$ is a hyperparameter to balance these two loss terms ($\beta = 0.5$).

**Decoder model** The decoder trained via Eq. (13) could be a GNN or a MLP. Although the GNN decoder might be powerful (Hou et al., 2022), we are curious if the MLP delivers a comparable or better performance with higher efficiency.

## 6 Experiments

### 6.1 Experimental Settings

**Datasets** Following the setting of previous studies (Hou et al., 2022; Kim et al., 2022; Xia et al., 2023), 2 million unlabeled molecules from the ZINC15 dataset (Sterling & Irwin, 2015) was used to pre-train the GNN models. To evaluate the performance on downstream tasks, experiments were conducted across eight binary classification benchmark datasets from MoleculeNet (Wu et al., 2017).

Table 1: Test AUC (%) performance on eight molecular datasets comparing our method with baselines. The best AUC-ROC values for each dataset are in **bold**. All models use same GNN architecture except those indicated by *.

| | MUV | ClinTox | SIDER | HIV | Tox21 | BACE | ToxCast | BBBP | Avg |
|---|---|---|---|---|---|---|---|---|---|
| No Pretrain | $70.7_{\pm1.8}$ | $58.4_{\pm6.4}$ | $58.2_{\pm1.7}$ | $75.5_{\pm0.8}$ | $74.6_{\pm0.4}$ | $72.4_{\pm3.8}$ | $61.7_{\pm0.5}$ | $65.7_{\pm3.3}$ | 67.2 |
| MCM* Wang et al. (2022) | $74.4_{\pm0.6}$ | $64.7_{\pm0.5}$ | $62.3_{\pm0.9}$ | $72.7_{\pm0.3}$ | $74.4_{\pm0.1}$ | $79.5_{\pm1.3}$ | $61.0_{\pm0.4}$ | $71.6_{\pm0.6}$ | 69.7 |
| MGSSL Zhang et al. (2021) | $77.6_{\pm0.4}$ | $77.1_{\pm4.5}$ | $61.6_{\pm1.0}$ | $75.8_{\pm0.4}$ | $75.2_{\pm0.6}$ | $78.8_{\pm0.9}$ | $63.3_{\pm0.5}$ | $68.8_{\pm0.9}$ | 72.3 |
| Grover* Rong et al. (2020) | $50.6_{\pm0.4}$ | $75.4_{\pm8.6}$ | $57.1_{\pm1.6}$ | $67.1_{\pm0.3}$ | $76.3_{\pm0.6}$ | $79.5_{\pm1.1}$ | $63.4_{\pm0.6}$ | $68.0_{\pm1.5}$ | 67.2 |
| AttrMask Hu et al. (2020a) | $75.8_{\pm1.0}$ | $73.5_{\pm4.3}$ | $60.5_{\pm0.9}$ | $75.3_{\pm1.5}$ | $75.1_{\pm0.9}$ | $77.8_{\pm1.8}$ | $63.3_{\pm0.6}$ | $65.2_{\pm1.4}$ | 70.8 |
| ContextPred Hu et al. (2020a) | $72.5_{\pm1.5}$ | $74.0_{\pm3.4}$ | $59.7_{\pm1.8}$ | $75.6_{\pm1.0}$ | $73.6_{\pm0.3}$ | $78.8_{\pm1.2}$ | $62.6_{\pm0.6}$ | $70.6_{\pm1.5}$ | 70.9 |
| GraphMAE Hou et al. (2022) | $76.3_{\pm2.4}$ | $82.3_{\pm1.2}$ | $60.3_{\pm1.1}$ | $77.2_{\pm1.0}$ | $75.5_{\pm0.6}$ | $83.1_{\pm0.9}$ | $64.1_{\pm0.3}$ | $72.0_{\pm0.6}$ | 73.9 |
| Mole-BERT Xia et al. (2023) | $78.6_{\pm1.8}$ | $78.9_{\pm3.0}$ | $62.8_{\pm1.1}$ | $78.2_{\pm0.8}$ | $\mathbf{76.8}_{\pm0.5}$ | $80.8_{\pm1.4}$ | $64.3_{\pm0.2}$ | $71.9_{\pm1.6}$ | 74.0 |
| JOAO You et al. (2021) | $76.9_{\pm0.7}$ | $66.6_{\pm3.1}$ | $60.4_{\pm1.5}$ | $76.9_{\pm0.7}$ | $74.8_{\pm0.6}$ | $73.2_{\pm1.6}$ | $62.8_{\pm0.7}$ | $66.4_{\pm1.0}$ | 71.1 |
| GraphLoG Xu et al. (2021) | $76.0_{\pm1.1}$ | $76.7_{\pm3.3}$ | $61.2_{\pm1.1}$ | $77.8_{\pm0.8}$ | $75.7_{\pm0.5}$ | $83.5_{\pm1.2}$ | $63.5_{\pm0.7}$ | $72.5_{\pm0.8}$ | 73.4 |
| D-SLA Kim et al. (2022) | $76.6_{\pm0.9}$ | $80.2_{\pm1.5}$ | $60.2_{\pm1.1}$ | $78.6_{\pm0.4}$ | $\mathbf{76.8}_{\pm0.5}$ | $83.8_{\pm1.0}$ | $64.2_{\pm0.5}$ | $72.6_{\pm0.8}$ | 73.9 |
| **MoAMa** w/o $\mathcal{L}_{aux}$ | $78.5_{\pm0.4}$ | $84.2_{\pm0.8}$ | $61.2_{\pm0.2}$ | $\mathbf{79.5}_{\pm0.5}$ | $76.2_{\pm0.3}$ | $\mathbf{84.1}_{\pm0.2}$ | $\mathbf{64.6}_{\pm0.1}$ | $71.8_{\pm0.7}$ | 75.0 |
| **MoAMa** | $\mathbf{80.0}_{\pm0.8}$ | $\mathbf{85.3}_{\pm2.2}$ | $\mathbf{64.6}_{\pm0.5}$ | $79.3_{\pm0.6}$ | $76.5_{\pm0.1}$ | $80.1_{\pm0.5}$ | $63.0_{\pm0.4}$ | $\mathbf{72.8}_{\pm0.9}$ | **75.3** |

**Validation methods and evaluation metrics**    In accordance with previous work, we adopt a scaffold splitting approach (Hu et al., 2020a; Zhang et al., 2021). Random splitting may not reflect the actual use case, so molecules are divided according to structures into train, validation, and test sets (Wu et al., 2017), using a 80:10:10 split for the three sets. We use the area under the ROC curve (AUC) to evaluate the test performance of the best validation step during 10 independent runs.

**Model configurations**    For fair comparison with previous work, a five-layer Graph Isomorphism Network (GIN) with an embedding dimension of 300 was chosen for the GNN encoder. The READOUT strategy is mean pooling. During pre-training and fine-tuning, models were trained for less than 100 epochs using the Adam optimizer and a learning rate of 0.001. The batch sizes for pre-training and fine-tuning are 256 and 32, respectively.

## 6.2    BASELINES

There are two general types of baseline graph pre-training strategies that we evaluate our work against: **contrastive learning** tasks, such as D-SLA (Kim et al., 2022), GraphLoG (Xu et al., 2021), and JOAO (You et al., 2021), and **attribute reconstruction**, including Grover (Rong et al., 2020), AttrMask (Hu et al., 2020a), ContextPred (Hu et al., 2020a), GraphMAE (Hou et al., 2022), and Mole-BERT (Xia et al., 2023). Additionally, we evaluate on **motif-based pre-training** strategies, MGSSL (Zhang et al., 2021), which recurrently generates the motif tree for any molecule, and MCM (Wang et al., 2022), which uses a motif-based convolution module to generate embeddings.

## 6.3    RESULTS

We report AUC-ROC of different graph pre-training methods in Table 1. MoAMa outperforms all baseline methods on five out of eight datasets. On average, MoAMa outperforms the best baseline method Mole-BERT (Xia et al., 2023) by $1.3\%$ and the best contrastive learning methods D-SLA (Kim et al., 2022) by $1.4\%$. Even without the auxiliary loss $\mathcal{L}_{aux}$, our motif-aware masking strategy still maintains a performance improvement of $1.0\%$, which is still competitive with previous methods.

## 6.4    ABLATION STUDIES

To verify motif-aware masking parameters, we conduct ablation studies on the selection of masking distributions, reconstruction target attribute(s), reconstruction loss function, and decoder model.

**Study on Masking Distributions**    For motif-aware masking, there is the choice of masking the features of all nodes within the motif or choosing to only mask the features of a percentage of nodes within each sampled motif. For our study, we choose a motif coverage parameter to decide what percentage of nodes within each motif to mask, ranging from $25\%$, $50\%$, $75\%$, or $100\%$.

Furthermore, the masking strategy utilized by previous work performs node-wise masking (Hu et al., 2020a; Hou et al., 2022), where all features of a node are masked. An alternative strategy may be element-wise masking, where masked elements are chosen over all feature dimensions and implies that not all features of a node may necessarily be masked. Note that 100% masking will behave the exact same as node-wise masking, as 100% of nodes within a motif will have each feature masked.

Table 2: Strategy design for motif-aware attribute masking: (1) masking distribution, (2) reconstruction target, (3) reconstruction loss, and (4) decoder model. The chosen design is  highlighted .

| Design Space | | MUV | ClinTox | SIDER | HIV | Tox21 | BACE | ToxCast | BBBP | Avg |
|---|---|---|---|---|---|---|---|---|---|---|
| (1) | 100% Motif Coverage | $80.0_{\pm 0.8}$ | $85.3_{\pm 2.2}$ | $64.6_{\pm 0.5}$ | $79.3_{\pm 0.6}$ | $76.5_{\pm 0.1}$ | $80.1_{\pm 0.5}$ | $63.0_{\pm 0.4}$ | $72.8_{\pm 0.9}$ | **75.3** |
| | 75% Node-wise | $74.9_{\pm 1.1}$ | $82.3_{\pm 0.4}$ | $60.1_{\pm 0.3}$ | $78.8_{\pm 0.9}$ | $76.1_{\pm 0.1}$ | $82.3_{\pm 0.4}$ | $63.4_{\pm 0.1}$ | $72.1_{\pm 1.0}$ | 73.7 |
| | 75% Element-wise | $74.8_{\pm 0.7}$ | $84.9_{\pm 1.0}$ | $58.7_{\pm 0.1}$ | $79.7_{\pm 0.7}$ | $75.6_{\pm 0.1}$ | $85.7_{\pm 0.4}$ | $63.4_{\pm 0.2}$ | $72.6_{\pm 0.4}$ | 74.4 |
| | 50% Node-wise | $76.6_{\pm 1.2}$ | $86.4_{\pm 0.6}$ | $58.3_{\pm 0.1}$ | $78.1_{\pm 0.3}$ | $75.1_{\pm 0.2}$ | $81.9_{\pm 0.3}$ | $64.6_{\pm 0.1}$ | $72.7_{\pm 0.1}$ | 74.2 |
| | 50% Element-wise | $73.9_{\pm 0.2}$ | $71.2_{\pm 4.0}$ | $61.2_{\pm 0.4}$ | $77.5_{\pm 0.8}$ | $74.9_{\pm 0.4}$ | $81.1_{\pm 0.7}$ | $62.5_{\pm 0.1}$ | $70.6_{\pm 1.8}$ | 71.6 |
| | 25% Node-wise | $76.6_{\pm 1.5}$ | $86.3_{\pm 0.7}$ | $62.4_{\pm 0.2}$ | $78.4_{\pm 0.2}$ | $75.9_{\pm 0.2}$ | $81.8_{\pm 0.1}$ | $65.1_{\pm 0.1}$ | $74.7_{\pm 0.2}$ | 75.1 |
| | 25% Element-wise | $75.2_{\pm 1.5}$ | $82.1_{\pm 0.4}$ | $58.3_{\pm 0.1}$ | $77.8_{\pm 1.5}$ | $75.5_{\pm 0.2}$ | $81.5_{\pm 0.2}$ | $63.1_{\pm 0.1}$ | $71.6_{\pm 0.3}$ | 73.1 |
| (2) | Atom Type | $80.0_{\pm 0.8}$ | $85.3_{\pm 2.2}$ | $64.6_{\pm 0.5}$ | $79.3_{\pm 0.6}$ | $76.5_{\pm 0.1}$ | $80.1_{\pm 0.5}$ | $63.0_{\pm 0.4}$ | $72.8_{\pm 0.9}$ | **75.3** |
| | Chirality | $76.3_{\pm 1.8}$ | $75.1_{\pm 0.9}$ | $59.8_{\pm 0.5}$ | $77.9_{\pm 0.1}$ | $76.6_{\pm 0.1}$ | $79.8_{\pm 0.5}$ | $63.8_{\pm 0.2}$ | $73.8_{\pm 0.7}$ | 72.9 |
| | Both w/ one decoder | $76.2_{\pm 1.4}$ | $74.4_{\pm 1.1}$ | $62.4_{\pm 0.9}$ | $78.2_{\pm 1.1}$ | $75.5_{\pm 0.6}$ | $82.1_{\pm 0.4}$ | $64.3_{\pm 0.2}$ | $72.9_{\pm 0.2}$ | 73.3 |
| | Both w/ two decoders | $75.9_{\pm 0.9}$ | $81.5_{\pm 0.1}$ | $60.5_{\pm 0.1}$ | $78.5_{\pm 0.9}$ | $75.8_{\pm 0.2}$ | $82.0_{\pm 1.0}$ | $63.7_{\pm 0.3}$ | $73.4_{\pm 0.3}$ | 73.9 |
| (3) | Scaled Cosine Error | $80.0_{\pm 0.8}$ | $85.3_{\pm 2.2}$ | $64.6_{\pm 0.5}$ | $79.3_{\pm 0.6}$ | $76.5_{\pm 0.1}$ | $80.1_{\pm 0.5}$ | $63.0_{\pm 0.4}$ | $72.8_{\pm 0.9}$ | **75.3** |
| | Cross Entropy | $78.8_{\pm 1.1}$ | $84.5_{\pm 0.7}$ | $65.4_{\pm 0.2}$ | $78.6_{\pm 0.4}$ | $76.3_{\pm 0.1}$ | $82.4_{\pm 0.2}$ | $62.9_{\pm 0.5}$ | $72.3_{\pm 0.2}$ | 75.1 |
| | Mean Squared Error | $80.0_{\pm 0.5}$ | $84.1_{\pm 1.4}$ | $64.6_{\pm 0.5}$ | $78.3_{\pm 0.4}$ | $76.8_{\pm 0.2}$ | $80.5_{\pm 0.6}$ | $62.8_{\pm 0.3}$ | $71.8_{\pm 0.6}$ | 74.9 |
| (4) | GNN decoder | $80.0_{\pm 0.8}$ | $85.3_{\pm 2.2}$ | $64.6_{\pm 0.5}$ | $79.3_{\pm 0.6}$ | $76.5_{\pm 0.1}$ | $80.1_{\pm 0.5}$ | $63.0_{\pm 0.4}$ | $72.8_{\pm 0.9}$ | **75.3** |
| | MLP decoder | $78.8_{\pm 0.5}$ | $85.2_{\pm 0.1}$ | $65.5_{\pm 0.3}$ | $78.1_{\pm 0.6}$ | $76.2_{\pm 0.2}$ | $82.1_{\pm 0.6}$ | $62.8_{\pm 0.8}$ | $71.7_{\pm 0.4}$ | 75.1 |

We provide the predictive performance within Table 2. The predictive performance for the node-wise masking outperforms the element-wise masking for both 25% and 50% node coverage. At 75% coverage, element-wise masking outperforms node-wise. However, the full coverage masking strategy outperforms all other masking strategies, due to the hardness of the pre-training task, which enables greater transfer of inter-motif knowledge.

**Study on Reconstruction Targets**    The choice of attributes to reconstruct for GNNs towards molecular property prediction has traditionally been atom type (Hu et al., 2020a; Hou et al., 2022). However, there are other choices for reconstruction that could be explored. We verify the choice of reconstruction attrbutes by comparing the performance of the baseline model against models trained by reconstructing only chirality, both atom type and chirality using two separate decoders, or both properties using one unified decoder. From Table 2, we note that predicting solely atom type yields the best pre-training results. The second best strategy was to predict both atom type and chirality using two decoders. In this case, the loss of the two decoders are independent, leading to the conclusion that the chirality prediction task is ill-suited to be the pre-training task. Because choice of chirality is limited to four extremely imbalanced outputs, the useful transferable knowledge may be significantly lesser than that of atom prediction, which, for the ZINC15 dataset, has nine types.

**Study on Reconstruction Loss Functions**    For the pretraining task, we have three choices of error functions to calculate training loss. A standard error function used for masked autoencoders within computer vision (He et al., 2022; Zhang et al., 2022; Germain et al., 2015) is the cross-entropy loss, whereas previous GNN solutions utilize mean squared error (MSE) (Hu et al., 2020b; Park et al., 2019; Salehi & Davulcu, 2019; Wang et al., 2017). GraphMAE (Hou et al., 2022) proposed that cosine error could mitigate sensitivity and selectivity issues:

$$\mathcal{L}_{\text{rec}} = \frac{1}{|\mathcal{V}_{\text{[MASK]}}|} \sum_{v \in \mathcal{V}_{\text{[MASK]}}} (1 - \frac{\mathbf{X}_v^T \mathbf{H}_v}{||\mathbf{X}_v|| \cdot ||\mathbf{H}_v||})^\gamma, \gamma \geq 1. \tag{15}$$

This equation is called the scaled cosine error (SCE). $\mathbf{H}$ are the reconstructed features, $\mathbf{X}$ are the ground-truth node features, and $\gamma$ is a scaling factor ($\gamma = 1$) We investigate the effect these different error functions have on downstream predictive performance in Table 2 and find that SCE outperforms CE and MSE, in accordance with previous work.

**Study on Decoder Model Choices**    We follow the GNN decoder settings from previous work (Hou et al., 2022) to conduct our study to determine which decoder leads to better downstream predictive performance. In Table 2, we show that our method outperforms the MLP-decoder strategy, which support previous work that show MLP-based decoders lead to reduced model expressiveness because of the inability of MLPs to utilize the high number of embedded features (Hou et al., 2022).

### 6.5    INTER-MOTIF INFLUENCE ANALYSIS

In Table 3, we report the two InfRatio and three MRR measurements for our model and several baselines. A higher influence ratio indicates that inter-motif nodes have a greater effect on the target

Table 3: Measurements of inter-motif knowledge transfer using pre-trained models. A higher ratio is preferred for the InfRatio measurements, and a lower score is preferred for the MRR measurements.

| Model | **Avg** Test AUC | InfRatio$_{node}$ ↑ | InfRatio$_{graph}$ ↑ | MRR$_{node}$ ↓ | MRR$_{graph}$ ↓ | MRR$_{motif}$ ↓ |
|---|---|---|---|---|---|---|
| AttrMask | 70.8 | 0.70 | 0.44 | 0.66 | 0.64 | 0.51 |
| MGSSL | 72.3 | 0.60 | 0.38 | 0.77 | 0.75 | 0.64 |
| GraphLoG | 73.4 | 0.79 | 0.50 | 0.61 | 0.59 | 0.48 |
| D-SLA | 73.8 | 0.76 | 0.49 | 0.67 | 0.66 | 0.44 |
| GraphMAE | 73.9 | 0.76 | 0.48 | 0.64 | 0.61 | 0.49 |
| Mole-BERT | 74.0 | 0.66 | 0.42 | 0.72 | 0.70 | 0.59 |
| MoAMa | **75.3** | **0.80** | **0.51** | **0.59** | **0.55** | **0.41** |

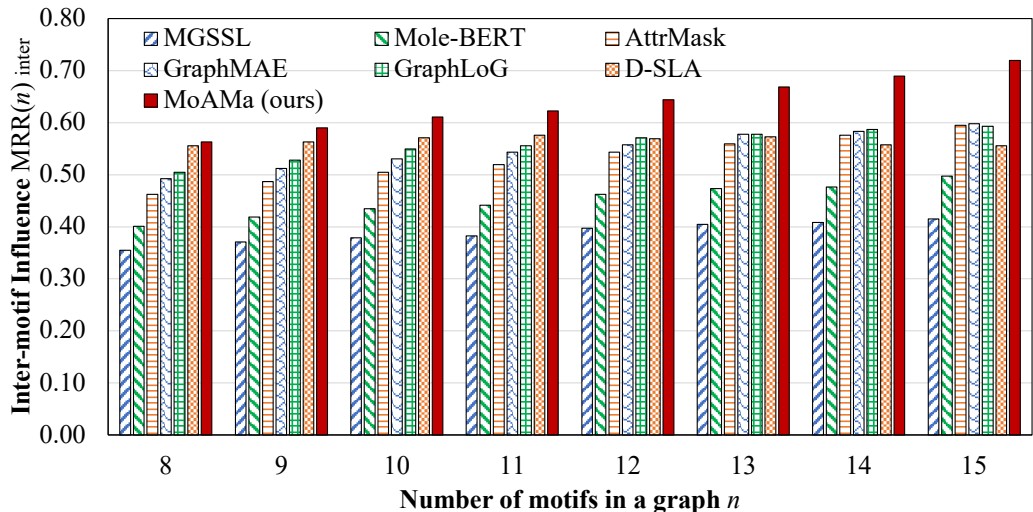

Figure 2: Inter-motif knowledge transfer score by motif count. A higher MRR$_{inter}^{(n)}$ score denotes greater inter-motif knowledge transfer.

node. The relatively low values indicate that the intra-motif node influence is still highly important for the pre-training task, but our method demostrates the highest inter-motif knowledge transfer amongst the baselines. We see that there is a small positive correlation between the average test AUC for each model and the InfRatio measurements, which supports our claim that greater inter-motif knowledge transfer leads to higher predictive performance. For the MRR measurements, our method boasts the lowest scores, which indicates less intra-motif knowledge dependence and greater inter-motif knowledge transfer.

For the sake of clear visualization, we define an inter-motif score which indicates inter-motif knowledge transfer according to the number of motifs $n$ within a graph:

$$\mathrm{MRR}_{\mathrm{inter}}^{(n)} = 1 - \frac{1}{\sum_{(\mathcal{V},\mathcal{E})\in\mathcal{G}^{(n)}} |\mathcal{V}|} \sum_{(\mathcal{V},\mathcal{E})\in\mathcal{G}^{(n)}} \sum_{v\in\mathcal{V}} \frac{1}{\mathrm{rank}_v}. \tag{16}$$

Figure 2 shows that our method outperforms all other models in terms of inter-motif knowledge transfer as shown by the higher MRR$_{inter}^{(n)}$ scores across different motif counts. Additionally, the inter-motif knowledge transfer using our method becomes more pronounced on graphs with higher numbers of motifs.

## 7 CONCLUSIONS

In this work, we introduced a novel motif-aware attribute masking strategy for attribute reconstruction during graph model pre-training. This motif-aware masking strategy outperformed existing methods that used random attribute masking, and achieved competitive results with the state-of-the-art methods because of the explicit transfer of long-range inter-motif knowledge and intra-motif structural information. We quantitatively verify the increase in inter-motif knowledge transfer of our strategy over previous works using inter-motif node influence measurements.

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

Table 4: Test RSME performance on three molecular datasets comparing our method with baselines. The best RMSE values for each dataset are in **bold**.

| | ESOL | FreeSolv | Lipophilicity | **Avg** |
|---|---|---|---|---|
| No Pretrain | $1.388_{\pm 0.05}$ | $2.965_{\pm 0.14}$ | $0.794_{\pm 0.005}$ | $1.716_{\pm 0.065}$ |
| MGSSL Zhang et al. (2021) | $1.259_{\pm 0.01}$ | $2.519_{\pm 0.006}$ | $\mathbf{0.722}_{\pm 0.002}$ | $1.500_{\pm 0.006}$ |
| AttrMask Hu et al. (2020a) | $1.307_{\pm 0.01}$ | $2.699_{\pm 0.03}$ | $0.766_{\pm 0.002}$ | $1.591_{\pm 0.014}$ |
| ContextPred Hu et al. (2020a) | $1.350_{\pm 0.08}$ | $2.784_{\pm 0.06}$ | $0.777_{\pm 0.008}$ | $1.637_{\pm 0.049}$ |
| GraphMAE Hou et al. (2022) | $1.235_{\pm 0.05}$ | $2.598_{\pm 0.02}$ | $0.755_{\pm 0.008}$ | $1.530_{\pm 0.026}$ |
| Mole-BERT Xia et al. (2023) | $1.239_{\pm 0.02}$ | $\mathbf{2.504}_{\pm 0.03}$ | $0.740_{\pm 0.006}$ | $\mathbf{1.494}_{\pm 0.019}$ |
| JOAO You et al. (2021) | $1.341_{\pm 0.02}$ | $3.243_{\pm 0.09}$ | $0.774_{\pm 0.008}$ | $1.786_{\pm 0.039}$ |
| GraphLoG Xu et al. (2021) | $1.341_{\pm 0.01}$ | $2.742_{\pm 0.01}$ | $0.739_{\pm 0.008}$ | $1.607_{\pm 0.009}$ |
| D-SLA Kim et al. (2022) | $1.289_{\pm 0.02}$ | $2.526_{\pm 0.01}$ | $0.730_{\pm 0.004}$ | $1.515_{\pm 0.011}$ |
| **MoAMa** | $\mathbf{1.228}_{\pm 0.01}$ | $2.552_{\pm 0.01}$ | $0.746_{\pm 0.001}$ | $1.509_{\pm 0.007}$ |

## A  APPENDIX

### A.1  INTER-MOTIF INFLUENCE EVALUATION COMPLEXITY

In the worst-case, evaluation of inter-motif node influence can be computed between every pair of nodes within a molecule, causing an evaluation complexity of $O(n^2)$, for $n$ is the number of nodes in a graph $G$. However, GNN message passing is limited by the number of layers used, $k$. Therefore, the node influence calculations will only need to be performed on neighbors within a $k$-hop radius of each other. This means that the time complexity of our evaluation is $O(n\bar{d}^k)$, where n is the number of nodes in the graph, $k$ is the number of layers of our GNN ($k = 5$), and $\bar{d}$ is the average degree of a node. $\bar{d}^k \leq n$ as molecular graphs are sparse, so the evaluation is not nearly as inefficient as $O(n^2)$.

### A.2  REGRESSION TASKS

We conducted additional evaluations on three regression datasets from MoleculeNet, ESOL, FreeSolv, and Lipophilicty (Wu et al., 2017). We use RMSE to measure the test performance of the best validation step during 3 independent runs.

Our method outperforms all baselines on the ESOL dataset and shows comparative results with previous methods when considering average RMSE across all three datasets.

### A.3  CASE STUDY

In Figure 3 there are two pairs of molecules, colored in blue and purple, that domain experts suggest to study. They are pairs because they look similar, have mostly similar properties, and have some different properties due to the structural differences. The molecular embedding space is obtained by a two-dimensional t-SNE algorithm on the pre-trained embeddings of the methods. The distributions and average distance between the graph examples are similar across the methods. However, contrastive learning strategies such as GraphLoG and JOAO failed to capture the proximities and put the pairs too distantly from each other. AttrMask based on random masking strategy was not able to learn from the structural difference at a higher level and put the pairs too close to each other. The proposed MoAMa provides a more reasonable set of embeddings for downstream fine-tuning.

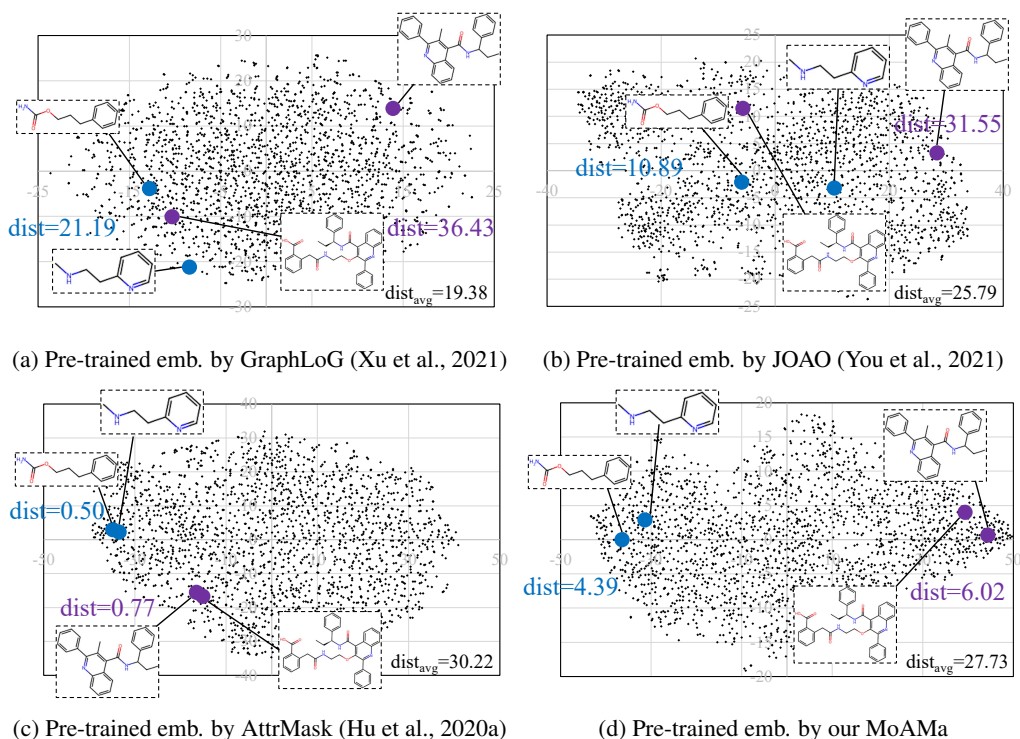

(a) Pre-trained emb. by GraphLoG (Xu et al., 2021)    (b) Pre-trained emb. by JOAO (You et al., 2021)

(c) Pre-trained emb. by AttrMask (Hu et al., 2020a)    (d) Pre-trained emb. by our MoAMa

Figure 3: Case study: MoAMa in (d) preserves the structural proximities of molecules better than other methods in (a-c) into pre-trained graph embeddings. The blue pair and purple pair are similar molecules yet have motif-based structural difference.

