# OpenReview forum: "Motif-aware Attribute Masking for Molecular Graph Pre-training"
_ICLR.cc/2024/Conference — Submitted to ICLR 2024_

### Official Review · Reviewer_PTf2 · 2023-10-28

**Soundness:** 2 fair
**Presentation:** 1 poor
**Contribution:** 2 fair
**Rating:** 3
**Confidence:** 4

**Summary:**

The paper introduces a masking strategy named MoAMa centered on chemical motifs. By breaking down each molecular graph into distinct motifs and masking node features within them, the method reconstructs those masked nodes. The paper also proposes an evaluation method considering inter-motif influence to help analyze the pre-training model on molecular dataset.

**Strengths:**

1. Pre-training for molecular representation learning is important.
2. Inter motif influence is a good aspect which needs to be further studied on pre-training.

**Weaknesses:**

1. My first question is that is inter-motif influence considered in the proposed method? In my understanding, inter-motif influence is only considered as an evaluation metric. Is it considered in the pre-training process?
2. If the inter-motif influence is an evaluation process, then the main contribution of the MoAMa is masking the whole motif instead of masking nodes. It comes up with an issue that completely masking the entire motif might be excessively difficult for the pre-trained GNN to reconstruct. Specifically, if all the nodes within a k-hop range of a motif are masked, then that particular node will never acquire any information about the features of its neighbors.
3. The complexity of calculating inter-motif influence needs to be discussed. For each graph, influence between any two nodes needs to be calculated by applying two modified graphs into a graph neural network, which means it needs to run the graph neural networks at most O(N^2) times. I highly recommend authors discuss the complexity of their proposed evaluation method.
4. How have the authors presented the test AUC? Is it the test AUC corresponding to the best validation AUC, or is it the final test AUC? It's essential to provide clarity on these experimental configurations to ensure the results can be reproduced.
5. The layout of the paper, especially section 4.3, is unclear. I found it challenging to link section 4.3 with its preceding section. Additionally, section 4.4 appears to have a closer relation to section 4.2, given its extended discussion on the pre-training strategy.

**Questions:**

Please refer to Weaknesses section.

---

> ### Author Response · Authors · 2023-11-19
>
> Thank you to the reviewer for providing us with their comments and questions for the manuscript. Below we will address the previously listed concerns.
>
> 1. We use inter-motif influence to evaluate the long-range message passing of our model. The limitation of previous work is the over-reliance of a node’s direct neighbors during training and the propagation bottleneck. We demonstrate that our model, with no modification to the underlying GNN architecture, is able to better mitigate issues related to feature over-squashing. The pre-training process itself is agnostic to the inter-motif influence measurements, as we can only measure the influence on a pre-trained network.
>
> 2. We address this concern using our motif selection criteria in lines 153-154. For criteria 1, all nodes within the masked motif must be within a k-hop distance from a node with visible features. Therefore, it is guaranteed that all masked nodes will receive information on some of its neighbors (excluding intra-motif neighbors). This pre-training task is indeed more difficult, but not excessively difficult. This added challenge encourages the model to learn longer-range dependencies and node interactions. Additionally, in practice, the average motif has ~2.4 nodes. With a 5-layer GIN, the masked nodes will still receive sufficient knowledge of its inter-motif neighbors.
>
> 3. GNN message passing is limited by the number of layers used, k. Therefore, the node influence calculations will only need to be performed on neighbors within a k-hop radius of each other. This means that the time complexity of our evaluation is O(n * d^k), where n is the number of nodes in the graph, k is the number of layers of our GNN (k = 5), and d is the average degree of a node. d^k <= n as molecular graphs are sparse, so the evaluation is not nearly as inefficient as O(n^2). Additional related analysis can be found in this work (1).
>
> 4. Following the experimental setup of this pioneering graph attribute masking work (2), we report the test ROC-AUC at the best validation epoch.
>
> 5. Thank you for this comment! We have opted to move our discussion of inter-motif influence (Sec. 4.3) in order to preserve the logical flow Sec. 4.2 and 4.4.
>
> (1) Nikolentzos, Giannis, George Dasoulas, and Michalis Vazirgiannis. "k-hop graph neural networks." Neural Networks 130 (2020): 195-205.
>
> (2) Hu, Weihua, et al. "Strategies for pre-training graph neural networks." arXiv preprint arXiv:1905.12265 (2019).

---

> > ### Comment · Reviewer_PTf2 · 2023-11-20
> > **Response to Authors**
> >
> > I thank the authors for their clarification.
> > For the comments, I have some additional questions.
> > 1. For criteria 1, is there a possibility that the selection of motifs might exclude larger motifs? Given that fragment decomposition methods such as BRICS typically extract larger motifs, does this criterion potentially lead to omitting significant motifs?
> > 2. Given that the Inter-motif influence method represents a significant contribution to this paper, it is crucial for the authors to delve into and analyze its complexity in the main section.
> > 3. As the Inter-motif influence is not included in the Motif-aware Attribute Masking strategy, I highly suggest that the authors incorporate it in both the paper's abstract and the introduction section.

---

> > > ### Author Response · Authors · 2023-11-20
> > >
> > > Thank you again for your comments!
> > >
> > > 1. The possibility for larger motifs to be excluded from selection exists. However, from our statistical analysis on the ZINC15 dataset, only about 1.3% of motifs were excluded from selection. Furthermore, while these motifs may be excluded from masking selection, these motifs still play critical roles by passing feature signals to the masked inter-motif nodes. Should the limitations of GNNs be alleviated and the standard number of layers increase past k = 5, then it would be interesting to revisit this work and utilize 100% of motifs in the selection process.
> > >
> > > 2. & 3. Thank you for the suggestions for improving our paper! As inter-motif influence is an important aspect of our work, we will include details of the analysis complexity in the methods section and incorporate more discussion of our measurements in the abstract and introduction. We have briefly included reference to our new measurements in the introduction in lines 51-52, but we will update this section with more details for our final draft.

---

### Official Review · Reviewer_Kr46 · 2023-10-30

**Soundness:** 3 good
**Presentation:** 3 good
**Contribution:** 3 good
**Rating:** 6
**Confidence:** 4

**Summary:**

Attribute reconstruction in graph neural network pre-training can be used to predict properties of molecules. Previous methods mask out randomly chosen nodes and rely too much on local neighbors, hindering the model's ability to capture long-range dependencies. In this work, the authors introduce a motif-aware attribute masking strategy for graph model pre-training, outperforming random masking methods. It is explained that the proposed approach obtain the advantage by transferring long-range inter-motif knowledge and intra-motif structural information. Ablation studies were conducted to support the explanations.

**Strengths:**

The motif-aware attribute masking strategy graph model pre-training is new.
The manuscript is well written and is easy to follow.

**Weaknesses:**

The proposed approach was applied to binary classification problems. It will be interesting to see its performance on regression problem (e.g., the QM9 dataset)

**Questions:**

Experiments show that the proposed approach does not always outperform existing methods. It will be nice if the authors can provide some insights.

---

> ### Author Response · Authors · 2023-11-20
>
> Thank you for the suggestion! We have performed additional experiments on the Lipophilicity, ESOL, and FreeSolv datasets. Our method shows comparable average performance to previous methods and performs the best out of the previous methods on the ESOL dataset. We reported the test RMSE for the best validation RMSE across 3 independent runs. For brevity, we have included the average RMSE across all three datasets below.
>
> None: 1.716
>
> AttrMask: 1.591
>
> MGSSL: 1.500
>
> GraphLoG: 1.607
>
> JOAO: 1.786
>
> GraphMAE: 1.530
>
> D-SLA: 1.515
>
> Mole-BERT: 1.494
>
> MoAMa: 1.509
>
> For molecular property prediction, it has become standard practice to separate the training, validation, and test sets according to molecular scaffolds, i.e. fragments. Therefore, the data distribution on the training set will not reflect that of the validation or test sets. This procedure makes the focus of predictive models to be on generalization. However, when training, it is hard to know what knowledge will be most useful for generalization, especially when considering multiple downstream datasets. Hence, the trained knowledge from pre-training and the fine-tuning training set may not always be useful for evaluation on the validation and test sets. Our method shows great generalization on 5 of 8 downstream datasets, but the knowledge transferred for the remaining 3 may not be as useful as the knowledge transferred from other models.

---

### Official Review · Reviewer_871j · 2023-10-30

**Soundness:** 2 fair
**Presentation:** 3 good
**Contribution:** 2 fair
**Rating:** 5
**Confidence:** 4

**Summary:**

The paper proposes a motif-aware atom masking approach for pre-training on molecular data. The idea behind is that an atom embedding should not only be influences by the closest neighbor nodes, and the paper includes some more formal analysis about how the proposed masking may adapt node influence scores. The experiments compare to other works on the moleculenet benchmark and contain a series of ablation studies.

**Strengths:**

- I agree that masking is under-investigated in molecular SSL.
- The proposed approach is straightforward and makes sense to me, and the analysis in terms of node influence is nice.
- The ablation experiments offer some more insights, although some (e.g., studies on loss, MLP projection) seem to me rather side information not exactly supporting the main contribution.

**Weaknesses:**

- The related work is missing all references to masking approaches in SSL beyond graphs. ICLR is a more general DL conference and graph SSL has clearly been inspired by those, and there are various works on masking, including theory.
- Some strong assumptions of the proposed approach are not enough backed by references, in my opinion. And the experiment design largely makes use of them (the proposed MRR metrics, ablation studies).
   * l. 100: "For molecular graphs, random attribute masking results in either over-reliance on intra-motif neighbors or breaking the inter-motif connections via random edge masking." - may need some more explanation and references that this really hurts the embedding
   * l. 35: "The presence and interactions between chemical motifs directly influence molecular properties, such as reactivity and solubility (Frechet, 1994; Plaza et al., 2014). Therefore, to capture the interaction information between motifs, it is important to transfer inter-motif structural knowledge and other long-range dependencies during the pre-training of graph neural networks. - while this says that interactions are important, it does not say that neighbor information is not important or, specifically, that it is less important
   * "Unfortunately, the random attribute masking strategies used in previous work for graph pre-training were not able to capture the long-range dependencies inherent in inter-motif knowledge (Kipf & Welling, 2016; Hu et al., 2020b; Pan et al., 2019). That is because they rely on neighboring node feature information for reconstruction (Hu et al., 2020a; Hou et al., 2022)." - Usually, GIN is applied with 5 layers and the studied molecules are rather small.

- Sec 5.3
   * l.271 "MoAMa outperforms all previous methods" - maybe "all we report here"
   * While I think that this type of inter-motif information is relevant, I think the experiments should focus on showing that it is complementary or improving upon regular masking. However, this is not visible from the experiments reported. Also, the paper could make more visible which models are directly comparable in terms of masking (e.g., since they use the same baseline architecture and pre-training data) and which not (e.g., the comparison to Grover is lacking since the baseline model is already very different).
   * It seems, for a comparison of the masking approach itself, the focus has to be on the "w/o L_aux" model, howver, then the performance increases compared to the baselines are not as clearly showing that this inter-motif information is really the best one.

**Questions:**

See above.

---

> ### Author Response · Authors · 2023-11-19
>
> Thank you to the reviewer for their comments on how we can improve our manuscript! Below we provide our response to the comments raised above.
>
> 1. We have briefly mentioned some related works within NLP and CV that utilize semantically related regions for masking (lines 29-32, 53-57), however we agree that a more detailed discussion would be important to contextualize this work to the larger research community. We will include a more detailed discussion of related masking works, such as Masked Language Modeling and Masked Autoencoding in CV in our final version.
>
> 2.1 Graph transformers, which have been growing in popularity, have been shown to outperform traditional GNNs on different graph prediction tasks, including those in the material science domain (1, 2, 3). The strength of these transformer models is the leveraging of full-connections among nodes in the graph, allowing the capture of long-range dependencies. Therefore, long-range dependencies are shown to increase performance of prediction models. However, many message passing GNNs struggle to pass long-range information, even when considering the k-hop neighborhood restriction. Many MPNN perform better when only considering a node’s 1-hop neighborhood (4) due to the propagation bottleneck and subsequent over-smoothing of features, even though long distance interactions have been shown to be important during training in chemical domains (5). Through our experiments, we have empirically measured the reliance of inter-motif neighbors on previous GNN based attribute masking models, and we have found that our method is better able to mitigate the propagation bottleneck on a 5-layer network while demonstrating notable downstream predictive results.
>
> 2.2 Both inter-motif and intra-motif interaction information are useful for downstream prediction. However, prior work in message passing for quantum chemistry has shown that long-range dependencies are important for downstream prediction in chemical domains (5). Additionally, traditional GNNs have the same expressiveness as the 1-WL test (6). In fact, generationalization to higher order graph structures is an on-going problem. One recent work proposes a message passing strategy that aggregates subgraph features during each message passing iteration (7), moving the expressivity boundary to the 3-WL test. Our work does not make any claims to extend GNNs expressivity beyond the 1-WL test, but we do leverage node features in a strategic way to overcome the propagation bottleneck. With smarter aggregation of features during each 1-hop step.
>
> 2.3 Even with only 5 layers, the bottleneck problem can still be observed (8). So even with small molecules, it is possible for over-squashing to remove long-range interactions during propagation. As shown through our experiments in Table 3, our method transferred the most inter-motif knowledge, thus mitigating the over-squashing problem more than previous works.
>
> 3.1 Thank you for this comment. We will modify our sentence to be more precise. We took care in selecting these SSL pre-training models because they were highly related to our work and represented SOTA in a diverse range of strategies.
>
> 3.2 In Table 1, we demonstrate that MoAMa outperforms previous masking works (Grover, AttrMask, ContextPred, GraphMAE, Mole-BERT). We do not believe that our method is complementary to regular masking as it replaces regular random masking with a motif-aware masking strategy. We will make a clearer distinction between which models are directly comparable (e.g. have the same pre-training data and architecture). All models except MCM and Grover utilize a 5-layer GIN with the ZINC15 dataset for pre-training.
>
> 3.3 Our model, even without L_aux, still holds a 1.0% average performance increase as compared to previous methods, which we believe is significant and represents SOTA.
>
> (1) Devin Kreuzer, Dominique Beaini, Will Hamilton, Vincent Létourneau, and Prudencio Tossou. Rethinking graph transformers with spectral attention.
> (2) Grégoire Mialon, Dexiong Chen, Margot Selosse, and Julien Mairal. GraphiT: Encoding graph structure in transformers.
> Chengxuan Ying, Tianle Cai, Shengjie Luo, Shuxin Zheng, Guolin Ke, Di He, Yanming Shen, and Tie-Yan Liu. Do transformers really perform badly for graph representation?
> (4) Dwivedi, Vijay Prakash, et al. "Long range graph benchmark."
> (5) Justin Gilmer, Samuel S Schoenholz, Patrick F Riley, Oriol Vinyals, and George E Dahl. Neural message passing for quantum chemistry.
> (6) Christopher Morris, Martin Ritzert, Matthias Fey, William L Hamilton, Jan Eric Lenssen, Gaurav Rattan, and Martin Grohe. Weisfeiler and leman go neural: Higher-order graph neural networks.
> (7) Feng, Jiarui, et al. "How powerful are k-hop message passing graph neural networks."
> (8) Alon, Uri, and Eran Yahav. "On the bottleneck of graph neural networks and its practical implications."

---

> ### Comment · Reviewer_871j · 2023-11-22
> **Reviewer Response**
>
> Thank you for the detailed responses, I'll consider updating my score after discussing with the other reviewers.
>
> 2 If the work focuses on long-range dependencies, then the experiments should maybe include experiments specifically comparing to "deeper" GNNs. My original comment was primarily focusing on the reasoning in terms of chemistry.
>
>
> 3.1 I think it would be indeed helpful to clarify to which of the common/existing masking strategies the paper compares to directly. The missing comparison to the Grover strategy is unlucky, but acceptable. In which terms is ContextPred a masking strategy?

---

> > ### Author Response · Authors · 2023-11-23
> >
> > 2. Deeper GNNs for use in molecular property prediction is an underexplored and potentially rich topic to investigate for future work. The main issue for deeper GNNs is the over-smoothing problem. Several recent works have attempted to minimize over-smoothing and utilize deeper GCNs for node classification. However, to the best of our knowledge, there are no works investigating deeper GNNs for graph classification tasks. This would indeed be a interesting and potentially fruitful future direction of study.
> >
> > Rusch, T. Konstantin, Michael M. Bronstein, and Siddhartha Mishra. "A survey on oversmoothing in graph neural networks." arXiv preprint arXiv:2303.10993 (2023).
> >
> > 3. We have made adjustments in the paper to clarify more specifically on which models are comparable architecturally. In the previous comment, a better term to use is "attribute reconstruction" rather than "masking". ContextPred predicts a surrounding graph structures based on the surrounding node's neighborhood, using the a similar reconstruction strategy as to AttrMask. The wording of our previous comment was ill-formed, but we point to Sec 5.2 where we classify ContextPred as an attribute reconstruction strategy.

---

### Official Review · Reviewer_sLns · 2023-10-31

**Soundness:** 4 excellent
**Presentation:** 3 good
**Contribution:** 3 good
**Rating:** 6
**Confidence:** 4

**Summary:**

This work proposes MoAMa to better enable the transfer of long-range inter-motif knowledge and intra-motif structural information by leveraging the information of atoms in neighboring motifs. Each molecule is decomposed into disjoint motifs, and the features for every node within a sample motif are masked. The graph decoder predicts the masked features of each node within the motif for reconstruction. Experimental results also prove the effectiveness of MoAMa.

**Strengths:**

(1)This paper is well-structured, logically sound, and provides a thorough explanation of MoAMa, with clear and concise figures.
(2)Due to the importance of chemical motifs for molecular properties and chemical reactions, the paper attempts to capture long-range dependencies from higher-level substructures, and the paper's motivation is sound and beneficial.
(3)The paper defines five inter-motif influence measurements to measure the inter-motif knowledge transfer of graph pre-training, and evaluates the models using these indicators, which is novel and significant.

**Weaknesses:**

(1)The paper only conducted experiments on one downstream task (classification). The main experimental results might not be quite sufficient.
(2)This paper lacks visualization results, case studies, and other interpretability analyses.

**Questions:**

(1)According to MGSSL[1], BRICS alone tends to generate motifs with large numbers of atoms. When the motif segmentation is too fine, many generated motifs are single atoms or bonds, which inhibits GNNs from learning higher-level semantic information through motif generation tasks. I wonder if the authors considered this issue when using only BRICS for motif fragmentation.
(2)The experimental datasets used in this study are all classification datasets in molecular property prediction. Could the authors validate regression datasets such as ESOL and FreeSolv, to further assess the effectiveness and generalization ability of MoAMa?
(3)I have some confusion about the experimental results for the baselines. The experimental results for the baselines in this paper show a certain discrepancy from the reported AUROC in the original baseline papers. Could the authors please clarify the reasons behind this difference?

[1] Zhang Z, Liu Q, Wang H, et al. Motif-based graph self-supervised learning for molecular property prediction[J]. Advances in Neural Information Processing Systems, 2021, 34: 15870-15882.

---

> ### Author Response · Authors · 2023-11-21
>
> Thank you to the reviewer for their positive comments for our work! We will address the weaknesses and questions below:
>
> 1. This was not an issue with our work. For MGSSL, the training process requires a full list of available motifs to generate, so it is vital to create an appropriate motif generation strategy that reduces the number of motif classes to generate, but does not reduce motifs down to single atoms. For our work, we are using motif segmentation to choose which nodes to mask for attribute masking. Therefore, the exact motif label itself is not necessary for training at all, only the segmentation. We considered the issue of finding appropriately sized motifs, but through statistical analysis, we found that motifs on average had 2.4 nodes each. Random attribute masking models the situation where each motif is of size 1, so already our method already takes into consideration higher-level semantic information during training as compared to the baseline AttrMask. Furthermore, we use the motif selection criteria to disallow the masking of nodes in motifs too great in size, written in lines 153-154. For these reasons we were able to use BRICS without encountering the same issues as the authors of MGSSL would have needed to deal with.
>
> 2. We have performed additional experiments on the Lipophilicity, ESOL, and FreeSolv datasets. Our method shows comparable average performance to previous methods and performs the best out of the previous methods on the ESOL dataset. We reported the test RMSE for the best validation RMSE across 3 independent runs. For brevity, we have included the average RMSE across all three datasets below.
>
> None: 1.716
>
> AttrMask: 1.591
>
> MGSSL: 1.500
>
> GraphLoG: 1.607
>
> JOAO: 1.786
>
> GraphMAE: 1.530
>
> D-SLA: 1.515
>
> Mole-BERT: 1.494
>
> MoAMa: 1.509
>
>
> 3. These results are compiled from these three recent works (1, 2, 3). If the same baseline was found in two or more works, the better result was reported. The experimental settings for these works were the same as ours, and to demonstrate the integrity of our work, we chose to use their reported numbers rather than our own. We did notice similar results in our own experiments when reproducing the results from the baselines, however.
>
>
> Lastly, to address the weaknesses brought up by the reviewer, we have conducted case studies on the embedding spaces of pairs of molecules with similar structures and but different properties. The results illuminate further why random masking cannot capture long-range signals effectively and map molecules with similar local structures to similar places in the embedding space. We will include these details in the appendix to our work.
>
>
> (1) Kim, Dongki, Jinheon Baek, and Sung Ju Hwang. "Graph self-supervised learning with accurate discrepancy learning." Advances in Neural Information Processing Systems 35 (2022): 14085-14098.
> (2) Xu, Minghao, et al. "Self-supervised graph-level representation learning with local and global structure." International Conference on Machine Learning. PMLR, 2021.
> (3) Jun Xia, Chengshuai Zhao, Bozhen Hu, Zhangyang Gao, Cheng Tan, Yue Liu, Siyuan Li, Stan Z. Li: Mole-BERT: Rethinking Pre-training Graph Neural Networks for Molecules. ICLR 2023

---

> > ### Comment · Reviewer_Kr46 · 2023-11-22
> > **More clarification needed**
> >
> > I am not sure how to digest the new information. Please elaborate a bit.
> > (a) How to read "Our method shows comparable average performance to previous methods and performs the best..." from the numbers listed?
> > (b) Would you please give more explanation to justify "The results illuminate further why random masking cannot capture long-range signals effectively and map molecules with similar local structures to similar places in the embedding space"?

---

> > > ### Author Response · Authors · 2023-11-23
> > >
> > > We apologize for any confusion. We have uploaded a revision of the manuscript that contains full details on the regression experiments (A.2) and the case study (A.3).

---

### Official Review · Reviewer_KumA · 2023-11-05

**Soundness:** 1 poor
**Presentation:** 2 fair
**Contribution:** 1 poor
**Rating:** 3
**Confidence:** 4

**Summary:**

The paper proposes a motif-based masking strategy for molecular representation pretraining. Unlike other studies that usually randomly mask node attributes, the paper proposes to mask the nodes in a motif manner, and then reconstruct them in the decoder. In such a way, the model will capture both intra-motif and inter-motif information. Furthermore, a knowledge-enhanced auxiliary loss is used to complement the reconstruction loss.

**Strengths:**

+ The paper is well-motivated, since random masking is the trend now. It is good to see that the authors advocate including domain knowledge such as motifs.
+ The ablation study is quite comprehensive, as it evaluates each component thoroughly.

**Weaknesses:**

- The proposed method does not seem solid enough. Many details lack theoretical support. For example, the criteria to restrain the masked motif selection is intuitive without much explanation. The authors claim that the proposed method is able to utilize graph structure, while it is unclear how the motif-aware masking addresses this issue. Another claimed contribution is to capture both local and long-range information, while it is not discussed how the motif masking captures long-range other than random masking. The masked motifs can still be close, and the message passing is still within the k-hop.
- The L_aux is proposed to complement L_rec since attribute masking focuses on local graph structures. Then this loss seems to fit with all the attribute masking methods. The results in Table 1 show that, MoAMa w/o Laux does not show much superiority compared with other baselines. It seems that the proposed motif-aware masking might perform limited. The authors should try Laux on other attribute masking methods to see if there are improvements. Also, it is not well elaborated on why and how Eq. 14 is designed.
- The results do not seem promising, and as the last point states, the effectiveness of the proposed motif-aware masking is questionable.
- The paper lacks an overall framework to help understand the detailed implementation.
- The technical novelty of the paper is relatively incremental for an ICLR paper. The paper primarily focuses on a narrow aspect of the pretraining method, switching from random masking to motif-wise random masking, which may have limited impact on the community.

**Questions:**

See weaknesses.

---

> ### Author Response · Authors · 2023-11-19
>
> Thank you to the reviewer for providing their insightful comments! We will provide our response to the concerns below.
>
> 1. The criteria on the masked motif selection are created directly due to the limitations of GNNs more broadly. Because of the over-smoothing issue of GNNs with many layers, GNNs are restricted in depth. Therefore, to guarantee that a masked node will receive feature signals from its inter-motif neighbors, we must restrict the size of the motifs being masked. We use criteria 1 to guarantee all nodes within a motif are passed relevant inter-motif knowledge. Criteria 2 is used to eliminate the cases where two large masked motifs are adjacent, and thus nodes near the boundary may not receive any propagation signals. Traditional GNNs have the same expressiveness as the 1-WL test and are thus unable to recognize larger order structures within graphs (1). Due to the propagation bottleneck inherent to GNNs, traditional GNN models of size k >= 4 are also unable to fully utilize local graph structures effectively (2). While staying within the constraints of the GNN architecture, our work seeks to mitigate the propagation bottleneck by encouraging the transfer of longer-range signals from a node’s inter-motif neighbors. With the retention of these longer range signals, our model is able to propagate featural and structural information further than other pretrained graph models. Because all nodes within a selected motif are masked, this inherently encourages the learning of local intra-motif structure. Additionally, our motif-aware masking strategy encourages the propagation of inter-motif signals. We develop inter-motif influence measurements in Sec 4.3 to demonstrate the weakness of previous methods and empirically show in Table 3 the increase of inter-motif knowledge transfer using our model as compared to baselines using the same GNN architecture.
>
> 2.1 Even without L_aux, our method is still competitive with a 1.0% average AUC-ROC performance increase across the baselines, which we believe still represents SOTA performance.
>
> 2.2 Thank you for suggesting this interesting direction for future work! Because the focus of this paper is on using a motif-aware masking strategy to encourage long-range signal transfer and mitigate over-squashing, we have chosen to leave this investigation of global motif knowledge for future work. With that being said, we have performed preliminary experiments using L_aux to complement the loss function of AttrMask and found a 5.6% performance increase from 77.8 to 83.4 on the BACE dataset, a 3.9% increase from 65.2 to 69.1 on the BBBP dataset, a 5.2% increase from 73.5 to 78.7 on the ClinTox dataset, and a 3.4% increase from 60.5 to 63.9 on the SIDER dataset. While our current strategy still outperforms AttrMask with L_aux, this increase in predictive performance is significant. Our L_aux certainly seems to complement this existing masking strategy well, and it would be a fruitful direction for further research.
>
> 2.3 Eq. 15 is inspired by eq. 5 from a similarity-based contrastive approach for 3D molecular representation learning (3).
>
> 3. See response to 2.1
>
> 4. In Figure 1, we represent the motif-aware masking strategy, in which the molecule is decomposed into disjoint motifs. Several motifs are selected according to masking criteria detailed in lines 153-154. For these masked nodes within the selected motifs, we used a graph autoencoder to reconstruct the graph node features, namely atom type, as our pre-training task. We have elucidated the design space of our method in Sec. 4.4 and Sec. 5.4 and provided the relevant experimental results, which should be comprehensive and insightful for understanding the graph autoencoder design.
>
> 5. We believe that this work not only contributes to the exploration of graph attribute reconstruction, but also provides a strong basis for future research in graph pre-training strategies. There are only a handful of fragment/motif based graph pretraining works, but the results from our work show that even a simple motif-based pretraining strategy is competitive with existing works. The SOTA results demonstrate that our simple yet effective method is able to utilize domain knowledge from motifs to enhance predictive performance over models which learn graph knowledge randomly. Because our work in motif-based graph pretraining mitigates feature over-squashing without the need to modify standard GNN architecture, we believe more sophisticated models can gain insights from our work to further improve GNN capabilities.
>
>
> (1) Christopher Morris, Martin Ritzert, Matthias Fey, William L Hamilton, Jan Eric Lenssen, Gaurav Rattan, and Martin Grohe. Weisfeiler and leman go neural: Higher-order graph neural networks.
> (2) Alon, Uri, and Eran Yahav. "On the bottleneck of graph neural networks and its practical implications."
> (3) Atsango, Austin, et al. "A 3D-Shape Similarity-based Contrastive Approach to Molecular Representation Learning."

---

> > ### Comment · Reviewer_KumA · 2023-11-22
> >
> > I appreciate the authors' efforts in addressing my concerns. However, I am not fully convinced by the rebuttal, especially for the method design and experimental results. The 2 criteria are assumed intuitively, but lack rigorous proof. Criteria #2 is "sampled motifs $\textbf{may  not}$ be adjacent". It seems that this criteria is conditional, the authors may provide more details about how they use these two criteria to establish the masking. A figure demonstration would help with elaborating on the implementation. As I mentioned initially, a comprehensive framework might also be helpful to better illustrate the proposed method. The current version lacks details, and the authors do not upload a revision to emphasize how they address all the reviewers' comments.
> >
> > In addition, for my concern regarding L_aux, the results are not impressive. The w/o L_aux even outperforms MoAMa on 3/8 datasets, and the overall improvement is only 75.0 -> 75.3, which is marginal. The effectiveness is doubtful.
> >
> >
> > p.s. Minor suggestion: The authors use "(1)" to represent the reference, which is a little bit confusing since normally such notation is used as bullet points to better summarize or categorize the context. "[1]" is commonly known as the formal reference format.

---

> > > ### Author Response · Authors · 2023-11-23
> > >
> > > As illustrated in Fig. 1, we must select non-adjacent motifs. For example, in the case where the three selected motifs are the leftmost 3 motifs, then the carbon atom on the left falls outside of the 5-hop neighborhood of the nearest non-masked node. This will result in the failure to transfer any meaningful inter-motif feature information to that node during message passing. The criteria are defined to choose motifs to mask so that no masked node is outside of the 5-hop neighborhood of any non-masked inter-motif node.
> > >
> > > We have made adjustments to the main body of the paper and included extra details in the appendix to address the reviewers concerns.

---

### Meta-Review · Area_Chair_Rk94 · 2023-12-09

**Metareview:**

This paper focuses on molecular graph pretraining use motif-aware attribute masking. While the motivation of using motif-aware masking is clear, reviewers still raised some critical concerns on 1) lack of important references, 2) limited novelty, 3) some results not convincing, etc. The authors tried to address these issues. However, the discussions did not help much to change the ratings. Considering that the majority of the reviewers gave negative scores after the rebuttal, my recommendation is to reject this paper.

**Justification For Why Not Higher Score:**

Reviewers still raised some critical concerns on 1) lack of important references, 2) limited novelty, 3) some results not convincing, etc. After the rebuttal, the scores from 5 reviewers are 3/3/5/6/6. The majority of the reviewers tend to reject the paper. Meanwhile, 2 remaining reviewers, who orignally gave positive scores, also did not champion this paper. Therefore we have to reject this paper.

**Justification For Why Not Lower Score:**

N/A.

---

### Decision · Program_Chairs · 2024-01-16

Reject